

# Increased number concentrations of small particles explains perceived stagnation in air quality over Korea

Sohee Joo[1], Juseon Shin[1], Matthias Tesche[2], Dehkhoda Naghmeh[1], Taegyeong Kim[1], Youngmin Noh[1]

[1]Division of Earth Environmental System Science, Pukyong National University, Busan 48513, Korea
[2]Leipzig Institute for Meteorology (LIM), Leipzig University, Leipzig, Germany

*Correspondence to:* Youngmin Noh (nym@pknu.ac.kr)

Abstract. The atmospheric visibility in South Korea has not improved despite decreasing concentrations of particulate matter (PM)$_{2.5.}$ Since visibility is influenced by particle size and composition as well as meteorological factors, the light detection and ranging data provided by the National Institute for Environmental Studies in Japan
and PM$_{2.5}$ measurements retrieved from Air Korea are used to determine the trends in PM$_{2.5}$ mass extinction efficiency (MEE) in Seoul and Ulsan, South Korea from 2015 to 2020. Moreover, the monthly trends in the Ångström exponent and relative and absolute humidity are determined to identify the factors influencing PM$_{2.5}$ MEE. The monthly average PM$_{2.5}$ MEE exhibits an increasing trend in Seoul [+0.04 (m$^2$·g$^{-1}$)·mth-1] and Ulsan [+0.07 (m$^2$·g$^{-1}$)·mth$^{-1}$]. Relative humidity increases by +0.070% and +0.095% per month in Seoul and Ulsan,
respectively, and absolute humidity increases by +0.029 and +0.010 (g·m$^{-3}$)·mth$^{-1}$, respectively. However, both Relative humidity and Absolute humidity are not statistically significant. The Ångström exponent increases by +0.005 and +0.011 per month in Seoul and Ulsan, respectively, indicating that the MEE increases as the particles become smaller each year. Such an increase in PM$_{2.5}$ MEE may have limited the improvements in visibility and adversely affected public perception of air quality improvement even though the PM$_{2.5}$ mass concentration in
South Korea is continuously decreasing.



## 1 Introduction

Particulate matter (PM)$_{2.5}$, which refers to particles with an aerodynamic diameter ≤2.5 μm, significantly affects atmospheric visibility, ecosystems, regional and global climate, and human health (Yue et al., 2017; De Marco et al., 2019; An et al., 2019; Li et al., 2019; Hao et al., 2021). As PM$_{2.5}$ is hazardous to human health, the International Agency for Research on Cancer (IARC), which is under the World Health Organization, classified PM$_{2.5}$ as Group 1 carcinogen in 2013 (IARC, 2013). The governments of South Korea and China, which are countries in Northeast Asia with severe air pollution due to fine particles, have also recognized the acute problem with PM$_{2.5}$ pollution and, thus, have enacted and implemented policies to address it (Jung, 2016; Van Donkelaar et al., 2016; Geng et al., 2019; Zhang and Geng, 2019; Zhai et al., 2019). In particular, the Chinese government implemented the Air Pollution Prevention and Control Action Plan in 2013 through strict emission controls (Gao et al., 2020); consequently, the concentration of PM$_{2.5}$ in China has decreased since 2013 (Zhang et al., 2019; Yue et al., 2020; Xiao et al., 2020; Liu et al., 2022), and such a reduction has also led to a decrease in the concentration of PM$_{2.5}$ in South Korea, which is located downwind of China (Xie and Liao, 2022). Similarly, the South Korean government has enacted various policies, including managing old diesel vehicles and expanding PM$_{2.5}$ monitoring stations. In fact, since the establishment of the nationwide PM$_{2.5}$ observation network and distribution of real-time observation data in 2015, the PM$_{2.5}$ concentration in South Korea has exhibited a decreasing trend (Bae et al., 2021).

However, despite the decreasing PM$_{2.5}$ concentration, public perception of fine particles has not significantly improved; several citizens still believe that PM$_{2.5}$ pollution remains serious. Among the environmental health issues included in the perception survey conducted by Won et al. (2022), experts perceived that climate change is the most critical issue; however, 80.9% of the citizens responded that PM$_{2.5}$ pollution is the most pressing environmental concern, confirming that most South Korean citizens are still afraid of PM$_{2.5}$ pollution.

Non-expert citizens perceive atmospheric PM$_{2.5}$ concentration visually: clear visibility indicates low PM$_{2.5}$ concentration and, thus, clean air; meanwhile, haziness indicates high PM$_{2.5}$ concentration. Visibility decreases as the concentration of fine particles suspended in the atmosphere increases because more particles scatter light (Fu-Qi et al., 2005; Liao et al., 2020). However, there have been several instances in Northeast Asia wherein improved visibility was not tantamount to reduced PM$_{2.5}$ concentration. Xu et al. (2020) reported that although the PM$_{2.5}$ levels in Guangzhou, China, decreased by more than 30% from 2013 to 2018, the frequency of low-visibility events only decreased by 5%. They suggested that the persistence of low visibility might prevent citizens from perceiving significant alleviation in aerosol pollution. Additionally, Liu et al. (2020) observed that although the PM$_{2.5}$ concentration in Eastern China continuously declined between 2013 and 2018, there was no corresponding improvement in visibility. They attributed this phenomenon to the increase in nitrate levels and relative humidity. Furthermore, Jeong et al. (2022) observed that the improvement in visibility in Seoul, South Korea, between 2012 and 2018 was not as pronounced as the decrease in PM$_{2.5}$ concentrations due to the composition of the particles.

Particle characteristics and meteorological conditions may alter the light-scattering properties of particles, which, in turn, affect visibility. When the size of particles in the atmosphere decreases, their light-scattering ability increases, leading to a reduction in visibility (Zhou et al., 2019). Furthermore, the extent to which particles scatter and absorb light varies depending on their composition (Yuan et al., 2006; Qu et al., 2015); for example, hydrophilic and hydrophobic particles differ in their light-scattering properties (Li et al., 2017). The aerosol mass extinction efficiency (MEE) is a key factor for assessing the degree of light scattering per unit mass, which varies



under different conditions. An increase in MEE indicates that although particles have the same mass concentration, changes in particle characteristics or meteorological conditions enhance the light-scattering ability of particles in the atmosphere.

Although several studies on MEE have been conducted in Northeast Asia, research on the underlying causes and long-term trends of MEE is scarce. Joo et al. (2022) used long-term visibility data in South Korea to calculate the

MEEs of $PM_{2.5}$ and $PM_{10}$, confirming an increasing trend in $PM_{2.5}$ and $PM_{10}$ MEEs compared with those in 2015. However, the accuracy of calculating the extinction coefficient of $PM_{2.5}$ using the Koschmieder formula, which involves subtracting the mass concentration values of $PM_{2.5-10}$ from visibility data, is limited because it may generate higher-than-the-actual values. Additionally, changes in particle size could not be confirmed, preventing the determination of the cause for the increase in MEE.

In this study, we use the light detection and ranging (lidar) data provided by the National Institute for Environmental Studies (NIES) of Japan through its Asian Dust and Aerosol Lidar Observation Network (AD-net) and the $PM_{2.5}$ mass concentration data provided by Air Korea to calculate the MEE of $PM_{2.5}$ and understand its trends. Section 2 describes the method of applying particle size information obtained from the lidar data to calculate the MEE of $PM_{2.5}$. In Section 3, we analyze the trends in $PM_{2.5}$ mass concentration, visibility, and $PM_{2.5}$

MEE variations; moreover, we examine the factors contributing to the variations in $PM_{2.5}$ MEE and the trends in $PM_{2.5}$ MEE in Northeast Asia. Finally, Section 4 summarizes the results of our research. As MEE is essential for converting the concentration obtained through optical measurements into the actual mass concentration of fine particles suspended in the atmosphere, the findings of this study contribute to the comprehensive assessment of the changes in PM characteristics in Northeast Asia.

**2 Methods**

**2.1 Analysis sites and data collection**

The AD-Net, operated by the NIES in Japan, is a lidar network that continuously monitors the vertical distribution of aerosols in Asia. Approximately 20 lidar observation stations have been instituted across Asia; currently, the observation stations in Korea are located in Seoul (37.46°N, 126.95°E), Ulsan (35.58°N, 129.19°E). The lidar

generates data for altitudes from 12 m to 17982 m at 30-m intervals every 15 min; moreover, it provides information on the mixing layer height, the aerosol depolarization ratio and backscatter coefficients at 532 and 1,064 nm wavelengths. A detailed description of the lidar system used to obtain the observation data can be found in the studies by Shimizu et al. (2016) and Xie et al. (2008). The lidar data recorded in Seoul and Ulsan from January 2015 to December 2020, which were obtained from the NIES website (http://www-lidar.nies.go.jp/AD-

Net/), are analyzed in this study.

To analyze the trend in PM mass concentration along with the optical concentration of fine patricles, data on atmospheric pollutants recorded by the monitoring stations near the lidar observation sites—Gwanak-gu, Seoul (37.49°N, 126.93°E) and Samnam-eup, Ulsan (35.56°N, 129.11°E)—are also analyzed. The monitoring stations in Seoul and Ulsan are respectively 3.77 and 7.57 km away from the designated lidar observation stations. The

mass concentration of fine particles is obtained from the final confirmed data on atmospheric pollutants available on the website of Air Korea (airkorea.or.kr), which is operated by the Korea Environment Corporation.



As visibility varies depending on the extent to which light is scattered or absorbed by particles or gases in the atmosphere and is correlated to the concentration of fine particles, it directly indicates the degree of air pollution (Huang et al., 2009; Shen et al., 2016). Accordingly, the visibility data recorded by the Meteorological

Administration for each city at the same time as the lidar measurements were taken are also analyzed. Additionally, to ensure the accuracy of the lidar data, ground meteorological observation data provided by the Meteorological Administration are used to exclude the lidar data recorded on rainy, snowy, or foggy days. Thus, only the PM concentration measurements and lidar data that are available at the same time are analyzed in this study. Moreover, to assess the impact of humidity on particles, the trends in relative humidity and absolute humidity are

simultaneously examined. Since the Korea Meteorological Administration currently only provides data on relative humidity, we use actual vapor pressure and temperature data to calculate the absolute humidity, which represents the ratio of water vapor mass to the total volume of air. The Vaisala formula is employed (Vaisala, 2013):

$AH = C \times (Ea \times 100)/(273.15 + T)$          Eq. (1)


where C represents the constant 2.16679 gK/J, Ea denotes the actual vapor pressure in hPa, and T represents the temperature in °C. Actual vapor pressure, temperature, visibility, and relative humidity data can be verified by referring to the ground meteorological observation data.

**2.2 Calculation of extinction coefficients for fine- and coarse-mode particles**

The extinction coefficient indicates the extent to which light is scattered and absorbed by particles and gases in the atmosphere. Generally, there is a positive correlation between the mass concentration and extinction coefficient of fine particles, with higher values observed on polluted days than clean-air days, leading to a reduction in visibility (Fu-Qi et al., 2005; Liao et al., 2020). Backscatter-related Ångström exponents (Å) are used

to distinguish between fine (index F) and coarse mode (index C) particles in the lidar measurements. The total backscatter coefficient ($\beta_T$) for each wavelength ($\lambda$) observed by lidar (532 and 1064 nm in our case) is equal to the sum of the backscatter coefficients of fine- ($\beta_F$) and coarse-mode particles ($\beta_C$):

$\beta_{T,\lambda} = \beta_{F,\lambda} + \beta_{C,\lambda}$          Eq. (2)


The Ångström exponent provides information about particle size with values closer to 0 indicating the dominance of large particles in the atmosphere (Ångström et al., 1929; Schuster et al., 2006). It is calculated as:

$Å_T = - \ln(\beta_{532}/\beta_{1064})/\ln(532/1064)$          Eq. (3)

Equation (3) can be expressed for the total backscatter coefficient, backscatter coefficient of fine-mode particles, and backscatter coefficient of coarse-mode particles, respectively.



$\beta_{T,532}/\beta_{T,1064} = e^{0.693\text{Å}_T}$      Eq. (4)

       $\beta_{F,532}/\beta_{F,1064} = e^{0.693\text{Å}_F}$      Eq. (5)

       $\beta_{C,532}/\beta_{C,1064} = e^{0.693\text{Å}_C}$      Eq. (6)

Substitution of Eqs. (5) and (6) into Eqs. (2) and (4), Eqs. (7) and (8) gives a way for deriving backscatter
coefficients of fine- and coarse-mode particles at 532 nm, respectively.

$$\beta_{F,532} = \frac{(e^{0.693(\text{Å}_T-\text{Å}_C)}-1)}{e^{0.693(\text{Å}_T-\text{Å}_C)}-e^{0.693(\text{Å}_T-\text{Å}_F)}}\beta_{T,532} \qquad \text{Eq. (7)}$$

$$\beta_{C,532} = \frac{(1-e^{0.693(\text{Å}_T-\text{Å}_C)})}{e^{0.693(\text{Å}_T-\text{Å}_C)}-e^{0.693(\text{Å}_T-\text{Å}_F)}}\beta_{T,532} \qquad \text{Eq. (8)}$$

PM concentration is referred to as $PM_{2.5}$, when the aerodynamic diameter of the particle is less than 2.5 μm and
as $PM_{10}$ when the aerodynamic diameter of the particle is less than 10 μm. In our analysis of aerosol optical
properties, the classification into fine mode and coarse mode assumes particles with an effective radius of less
than 1 μm as being in the fine mode, whereas particles with an effective radius larger than 1 μm are considered to
be in the coarse mode (Schuster et al., 2006; O'Neill et al., 2023). Consequently, the backscatter coefficients of
fine- ($\beta_{F,532}$) and coarse-mode particles ($\beta_{C,532}$) at 532 nm are assumed to correspond to the light extinction caused
by $PM_{2.5}$ and $PM_{2.5–10}$ particles, respectively. For the calculations, the Ångström exponent for fine-mode particles
($Å_F$) is assumed to be 3, while that for coarse-mode particles ($Å_C$) is assumed to be 0; such assumed Ångström
exponent values are calculated based on simulations using the Mie theory, considering particle volume size
distribution and a refractive index of 1.5 + 0.001i (Dubovik et al., 2002; Veselovskii et al., 2004).

By multiplying the lidar ratio (S) with β separated into fine- and coarse-mode particles, the final extinction
coefficients (α) for fine- and coarse-mode particles are calculated as :

     α    = S × β      Eq. (9)

In this study, lidar data are not reliable close to the surface. Consequently, the lidar signal data from 162 m to the
maximum height determined by the mixing layer height are averaged on an hourly basis for analysis.

### 2.3 LR and MEE

The lidar ratio varies depending on aerosol size, shape, type, composition, and other factors, it is important to
apply the appropriate LR to calculate the extinction coefficient (Kovalev and Eichinger, 2004; Cao et al., 2019).
In Korea, due to geographical characteristics, dust and pollution are often mixed and a wide variety of particle
types exist. Therefore, LR research has been conducted on various particle types (Noh et al., 2007; Noh et al.,
2008; Noh et al., 2011; Shin et al., 2015). We use the method described by Groß et al. (2011) for calculating LRs
when two particle types are mixed. The total LR and coarse-mode LR, which are calculated using this method,
are applied to the backscatter coefficients.



Depending on the fraction of dust, the extinction coefficients for a mixture of two aerosol types can be expressed as follows:

$$\alpha = \alpha_1 + \alpha_2 = (1-x)\alpha + x\alpha \qquad \text{Eq. (10)}$$

where $\alpha$ represents the total extinction coefficient, $\alpha_1$ is the extinction coefficient of Component1, $\alpha_2$ is the extinction coefficient due to Component2, and x can be expressed as $x = \alpha_1/\alpha$. To determine the proportion of dust, the dust ratio ($R_D$) is calculated using the depolarization ratio ($\delta_P$) at 532 nm, which represents the non-sphericity of particles, as shown in Eq. (11):

$$R_D = \frac{(\delta_P - \delta_2)(1 + \delta_1)}{(\delta_1 - \delta_2)(1 + \delta_P)} \qquad \text{Eq. (11)}$$

where $\delta_1$ and $\delta_2$ represent the depolarization ratios of pure dust particles and spherical aerosols, respectively. For the calculations in this study, $\delta_1$ and $\delta_2$ are set to 0.32 and 0.02, respectively, (Shimizu et al., 2004; Liu et al., 2008; Freudenthaler et al., 2009; Tesche et al., 2009; Nishizawa et al., 2011).


The LR for a mixture of two aerosol types can be expressed as follows:

$$S = \frac{1}{\frac{x}{S_1} + \frac{1-x}{S_2}} \qquad \text{Eq. (12)}$$

where $S_1$ and $S_2$ represent the LRs of the two aerosol components considered in a mixture; specifically, $S_1$ corresponds to dust, while $S_2$ corresponds to pollution particles within the total LR. x represents the proportion of

the total extinction coefficient accounted for by the extinction coefficient of Component1, which is dust. Furthermore, when calculating the coarse-mode LR, Component1 is set as dust, and Component2 is set as coarse-mode aerosol. In coarse mode LR, x represents the proportion of the coarse-mode particle extinction coefficient accounted for by the dust extinction coefficient of Component 1.

In this study, we established the range of LRs based on the research conducted by Shin et al. (2015) in Gwangju,

South Korea. Based on previous LR studies conducted in Korea, we assumed an LR ($S_1$) of 40 sr for dust and an LR ($S_2$) of 70 sr for pollution particles in this study. An LR ($S_1$) of 40 sr for dust and an LR ($S_2$) of 60 sr for coarse-mode particles are assumed and applied in the calculations (Fig. 1).





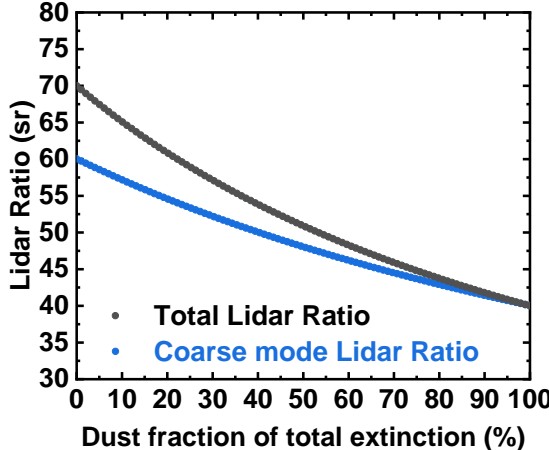

**Figure 1:** Total lidar ratio and coarse-mode lidar ratio as dust fraction of total extinction.


The calculated $R_D$ is inputted as the x value to determine the total extinction coefficient; the proportion of dust within coarse-mode particles is used to calculate the x value. The x value is applied to Eqs. (10) and (12) to calculate the total LR and coarse-mode LR, respectively. The calculated LRs are applied to the total backscatter

coefficients and coarse-mode backscatter coefficients to calculate the total extinction coefficients and coarse-mode extinction coefficients, respectively. Afterwards, the fine-mode extinction coefficients are obtained by subtracting the coarse-mode extinction coefficients from the total extinction coefficients. The total, coarse-mode, and fine-mode extinction coefficients are divided by the mass concentrations of $PM_{10}$, $PM_{2.5-10}$, and $PM_{2.5}$, respectively, to calculate the $MEE_{10}$, $MEE_{2.5-10}$, and $MEE_{2.5}$, respectively [Eq. (13)]:


$$MEE = \frac{\alpha}{PM} \qquad \text{Eq. (13)}$$

**2.4 Mann–Kendall test and Sen's slope**

The slope of all trend lines in this study is obtained through simple linear regression analysis. In addition, we

employ the Mann–Kendall test (MK test) and calculate Sen's slope to statistically confirm the accuracy of MEE trends (Mann, 1945; Kendall, 1957; Sen, 1968). The MK test is a non-parametric statistical test used to determine the occurrence of an increasing or decreasing trend in time series data; however, it does not provide information about the rate of change of the trend. By calculating the Sen's slope, it is possible to assume a linear trend in data and estimate the slope of the trend over time. The MK test is used to determine which between the null (no trend)

or alternative hypotheses (a clear trend) is appropriate. These hypotheses are determined based on z-scores and p-values. If |Z| is greater than 1.96, the alternative hypothesis is accepted at a 95% confidence level; however, if |Z| is greater than 2.57, the alternative hypothesis is accepted at a 99% confidence level. Additionally, the sign of the





z-scores can be used to determine an increasing or decreasing trend. A p-value below 0.05 is considered statistically significant, indicating a meaningful result.





## 3 Results and discussion

### 3.1 Visibility and PM concentration trends

Figure 2 and Table 1 show the monthly changes in PM$_{2.5}$, PM$_{2.5–10}$, and PM$_{10}$ concentrations with the visibility across Seoul and Ulsan consistent with the findings of previous studies, the monthly PM mass concentrations in both cities are higher in winter and spring, and lower in summer and autumn (Kim, 2020; Allabakash et al., 2022). The monthly PM$_{10}$ concentration exhibits a decreasing trend in both cities; the monthly PM$_{10}$ concentration decreases by $-0.31$(µg·m$^{-3}$)·mth$^{-1}$ in Seoul and $-0.21$ (µg·m$^{-3}$)·mth$^{-1}$ in Ulsan. However, the extent of the decrease in the monthly PM$_{2.5}$ and PM$_{2.5–10}$ concentrations in Seoul and Ulsan differ depending on particle size. The PM$_{2.5}$ concentration in Seoul decreases by $-0.11$ (µg·m$^{-3}$)·mth$^{-1}$, while the PM$_{2.5–10}$ concentration decreases by $-0.21$ (µg·m$^{-3}$)·mth$^{-1}$. The reduction in the PM mass concentration of coarse-mode particles (PM$_{2.5–10}$) is greater than that of fine-mode particles (PM$_{2.5}$). In Ulsan, the monthly PM$_{2.5}$ concentration decreases at a rate $[-0.12$ (µg·m$^{-3}$)·mth$^{-1}]$, and the PM$_{2.5–10}$ concentration decreases at a rate $[-0.08$ (µg·m$^{-3}$)·mth$^{-1}]$. However, the monthly visibility in Ulsan and Seoul does not change, showing $-0.00$ and $+0.00$ km·mth$^{-1}$, respectively. Through these monthly trends, it can be observed that despite the decrease in PM mass concentrations, the visibility in Seoul and Ulsan does not improve.

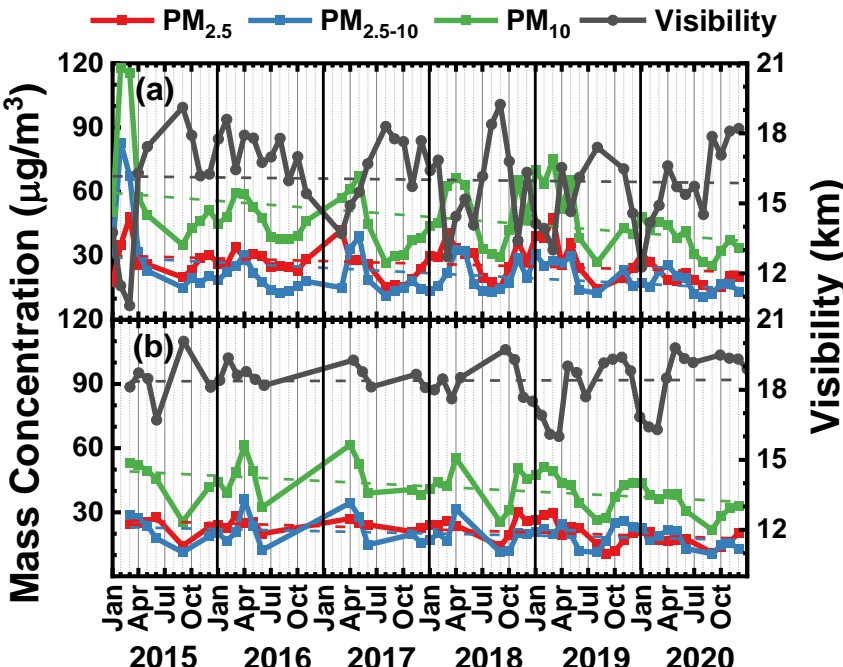

**Figure 2:** Monthly particulate matter (PM) mass concentration and visibility in (a) Seoul and (b) Ulsan.





These results can also be confirmed through the MK test (Table 1). In the case of $PM_{2.5}$, a decreasing trend is confirmed in both regions. The p-values in the $PM_{2.5}$ trends in both regions are all smaller than 0.05, indicating statistically significant decreasing trend. In the case of $PM_{2.5-10}$, Seoul shows a decreasing trend, unlike Ulsan. The z-scores of $PM_{2.5-10}$ in Seoul is -2.32, showing a confidence level of 95%. In $PM_{10}$, the z-scores of both regions are above 2.57, showing a decreasing trend at the 99% confidence level. However, despite the decreasing trend in PM mass concentration, statistical analysis does not reveal any significant trends in visibility for Seoul and Ulsan. The z-scores of the two regions are lower than 1.96, and the p-values are also greater than 0.05, showing no statistical significance.



**Table 1:** Mann–Kendall test and Slope for monthly PM mass concentration and Visibility.

|  |  | MK test |  |  | Slope |  |
| --- | --- | --- | --- | --- | --- | --- |
|  |  | Trend | z | P | Simple linear regression | Sen's slope |
| Seoul | $PM_{2.5}$ | Decreasing | -2.41 | 0.02 | -0.11 | -0.13 |
|  | $PM_{2.5-10}$ | Decreasing | -2.32 | 0.02 | -0.21 | -0.11 |
|  | $PM_{10}$ | Decreasing | -2.65 | 0.01 | -0.31 | -0.22 |
|  | Visibility | No trend | -0.96 | 0.34 | -0.00 | -0.01 |
| Ulsan | $PM_{2.5}$ | Decreasing | -3.71 | 0.00 | -0.12 | -0.20 |
|  | $PM_{2.5-10}$ | No trend | -1.67 | 0.10 | -0.08 | -0.11 |
|  | $PM_{10}$ | Decreasing | -3.41 | 0.00 | -0.21 | -0.37 |
|  | Visibility | No trend | +0.98 | 0.33 | +0.00 | +0.00 |

**3.2 MEE Trends**


Figure 3 and Table 2 show the monthly average MEE and the monthly MEE trend from 2015 to 2020. The $PM_{2.5}$ MEE in Seoul and Ulsan significantly increases by +0.04 and +0.07 ($m^2 \cdot g^{-1}$)·$mth^{-1}$, respectively. In contrast, the $PM_{2.5-10}$ MEE in Seoul and Ulsan exhibits a decreasing trend; it decreases by −0.02 and −0.03 ($m^2 \cdot g^{-1}$)·$mth^{-1}$, respectively. At the same PM mass concentration, the extent of light extinction by coarse-mode particles ($PM_{2.5-10}$) in Seoul and Ulsan steadily decreases, whereas the light extinction by fine-mode particles ($PM_{2.5}$) continues to

increase, as evidenced by the monthly average MEE trends.

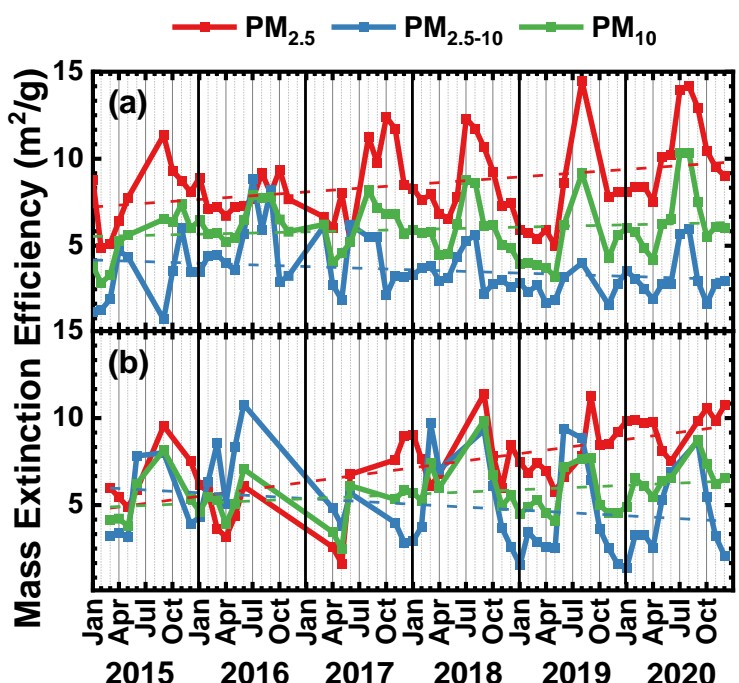

**Figure 3:** Monthly average mass extinction efficiency (MEE) in (a) Seoul and (b) Ulsan


The monthly average MEE trends based on particle size are also confirmed by the results of the MK test (Table 2). The z-scores of the $PM_{2.5}$ MEE in Seoul (+2.25) exceed 1.96, indicating an increasing trend at a 95% confidence level, and The z-scores of the $PM_{2.5}$ MEE in Ulsan (+4.81) exceed 2.57, indicating an increasing trend at a 99% confidence level; moreover, the p-values are all below 0.05, confirming the significance of the increasing

trend. The z-scores of the $PM_{2.5–10}$ MEE in Seoul are -2.14, indicating a decreasing trend at a 95% confidence level; additionally, the p-values are below 0.05, confirming the statistical significance of such a decreasing trend. However, The z-scores of the $PM_{2.5–10}$ MEE in Ulsan are −1.88, indicating that the z-score is below 1.96. It means that there is no statistically clear trend. The z-score of the $PM_{10}$ MEE in Seoul does not exhibit a distinct trend; however, the z-score and p-value of the $PM_{10}$ MEE in Ulsan are +2.21 and 0.03, respectively, indicating an

increasing trend at a 95% confidence level. Therefore, in both Seoul and Ulsan, the MEE of fine-mode particles ($PM_{2.5}$) exhibits a significant increasing trend, while that of coarse-mode particles ($PM_{2.5–10}$) in Seoul demonstrates a significant decreasing trend.

**Table 2:** Mann–Kendall test and Slope for monthly average mass extinction efficiency (MEE).

| MEE | MK-test | | | Slope | |
|---|---|---|---|---|---|
| | Trend | z | p | Simple linear regression | Sen's slope |





| | | | | | | |
|---|---|---|---|---|---|---|
| Seoul | PM$_{2.5}$ | Increasing | +2.25 | 0.02 | +0.04 | +0.03 |
| | PM$_{2.5–10}$ | Decreasing | -2.14 | 0.03 | -0.02 | -0.02 |
| | PM$_{10}$ | No trend | +0.35 | 0.73 | +0.01 | +0.00 |
| Ulsan | PM$_{2.5}$ | Increasing | +4.81 | 0.00 | +0.07 | +0.11 |
| | PM$_{2.5–10}$ | No trend | −1.88 | 0.06 | -0.03 | −0.03 |
| | PM$_{10}$ | Increasing | +2.21 | 0.03 | +0.02 | +0.03 |


As previously mentioned, the PM mass concentration of coarse-mode particles (PM$_{2.5–10}$) predominantly decreases in Seoul compared with that of fine-mode particles (PM$_{2.5}$); however, the visibility in Seoul does not improve[−0.00 km·mth$^{-1}$]. The light extinction in Seoul is predominantly caused by fine-mode particles (PM$_{2.5}$). Additionally, the monthly trend in PM$_{2.5}$ MEE in Seoul is +0.04 m$^2$·g$^{-1}$, indicating an increase in light extinction caused by fine particles, whereas the light extinction from PM$_{2.5–10}$ MEE exhibits a decreasing trend of −0.02 m$^2$·g$^{-1}$. Therefore, despite the overall decrease in PM mass concentration, the visibility in Seoul does not improve due to the high extinction efficiency and increasing trend of fine-mode particles. In Ulsan, although the change in PM$_{2.5–10}$ mass concentration is not significant [−0.08 (µg·m$^{-3}$)·mth$^{-1}$], the PM$_{2.5}$ mass concentration significantly decreases [−0.12 (µg·m$^{-3}$)·mth$^{-1}$], statistically indicating a reduction. However, the visibility in Ulsan does not improve, showing a rate of +0.00 km·mth$^{-1}$. Therefore, despite the decrease in PM$_{2.5}$ mass concentration, the significant monthly increase in PM$_{2.5}$ MEE [+0.07 (m$^2$·g$^{-1}$)·mth$^{-1}$] likely impacts the visibility in Ulsan. The observed monthly changes in MEE based on particle size in Seoul and Ulsan suggest that the characteristics of particles affecting light extinction vary on particle size. These changes are similar to the results found by Joo et al. (2022), who identified an increasing trend in PM$_{2.5}$ MEE in Seoul [+0.05 (m$^2$·g$^{-1}$)·mth$^{-1}$] through visibility data.


### 3.3 Analysis of the Causes of the Increasing PM2.5 MEE Trends

An increase in relative humidity and changes in particle size and composition contribute to the increase in MEE. An increase in relative humidity induces particle hygroscopic growth, thereby influencing the extinction coefficient (Zieger et al., 2011; Sabetghadam and Ahmadi-Givi, 2014; Qu et al., 2015; Dawson et al., 2020; Ting et al., 2022). Since a beta-radiation attenuation monitor (BAM) is used in South Korea to measure PM mass concentrations by removing humidity (Baek, 2022), the influence of increased relative humidity on PM mass concentration measurements may be less than that of the extinction coefficient, suggesting that the increase in relative humidity may influence the increase in MEE. For instance, Cheng-Cai1 et al. (2013) reported an increase in MEE with increasing relative humidity in Beijing, China, regardless of the season. Additionally, Liu et al. (2020) attributed the higher MEE in Eastern China in 2018 than that in 2013 to increased relative humidity and amount of nitrates.

Even with the same relative humidity, the extent to which aerosols absorb water varies depending on the size and composition of the particles; it can change the degree of light extinction, as particles grow larger than dry particles (Singh and Dey, 2012; Liu et al., 2013; Titos et al., 2016; Chen et al., 2019). Additionally, even with the same PM mass concentration in the atmosphere, a reduction in particle size can lead to an increased degree of light scattering, resulting in a larger MEE value (Zhou et al., 2019). Differences in hygroscopic growth are not only



influenced by particle size but also by particle composition, in which particles are classified into hydrophilic and hydrophobic species (Chen et al., 2014). Hydrophilic species include secondary inorganic compounds (e.g., sulfate, nitrate, and ammonium) and sea salt, while hydrophobic species include dust and black carbon.


In this study, changes in particle size and humidity are examined to determine the impact of humidity on MEE. Figure 4 and Table 3 show the monthly average trends in Ångström exponent, relative and absolute humidity in Seoul and Ulsan from 2015 to 2020. The monthly average trends in relative humidity in Seoul and Ulsan are +0.070 and +0.095 %·mth$^{-1}$, respectively. The relative humidity in Seoul and Ulsan exhibits an increasing trend,

although it is not pronounced. The monthly average absolute humidity in Seoul and Ulsan slightly increases by +0.029 and +0.010 (g·m$^{-3}$)·mth$^{-1}$, respectively. From the monthly change trends in relative and absolute humidity in both cities, no distinct trends are observed that could significantly impact MEE. In the MK test results shown in Table 3, the z-scores are all lower than 1.96, and the p-value is also higher than 0.05, indicating that the humidity in both regions does not have statistically significant trends.


To gain information about particle size, which can be another factor influencing MEE changes, Ångström exponent values are utilized (Ångström et al., 1929). Generally, a higher Ångström exponent value indicates that there are relatively more fine-mode particles in the atmosphere. Fu-Qi et al (2005) confirmed the effect of particle size on MEE by confirming that MEE increases as the Ångström exponent value increases. The Ångström

exponent in Seoul and Ulsan exhibits a clear increasing trend of +0.03 and +0.02 per month, respectively. The increasing trend of the Ångström exponent in Seoul and Ulsan is statistically significant according to the MK test (Table 3). The z-scores of the Ångström exponent in Seoul (+3.37) is greater than 2.57, indicating an increase in the Ångström exponent at a confidence level of over 99%. The z-scores of the Ångström exponent in Ulsan (+2.49) are greater than 1.96, indicating an increase in the Ångström exponent at a confidence level of over 95%. The p-

values of the Ångström exponent in Seoul and Ulsan are 0.00 and 0.01, respectively, which are lower than 0.05, confirming the statistical significance of the trend. In contrast, neither the z-scores nor the p-values of relative and absolute humidity are statistically significant; thus, the influence of humidity on the increasing or decreasing trend of MEE could not be confirmed. By using AERONET data, Lee and Bae (2021) and Shin et al. (2023) reported an increase in the Ångström exponent in Seoul, South Korea. Their results are consistent with the trends in the

Ångström exponent derived from the lidar measurements analyzed in this study. The increasing trend in the Ångström exponent in Seoul and Ulsan indicates a relative increase in the proportion of smaller particles compared to larger ones. It is believed that the Ångström exponent has influenced the increase in PM$_{2.5}$ MEE values in Seoul and Ulsan.



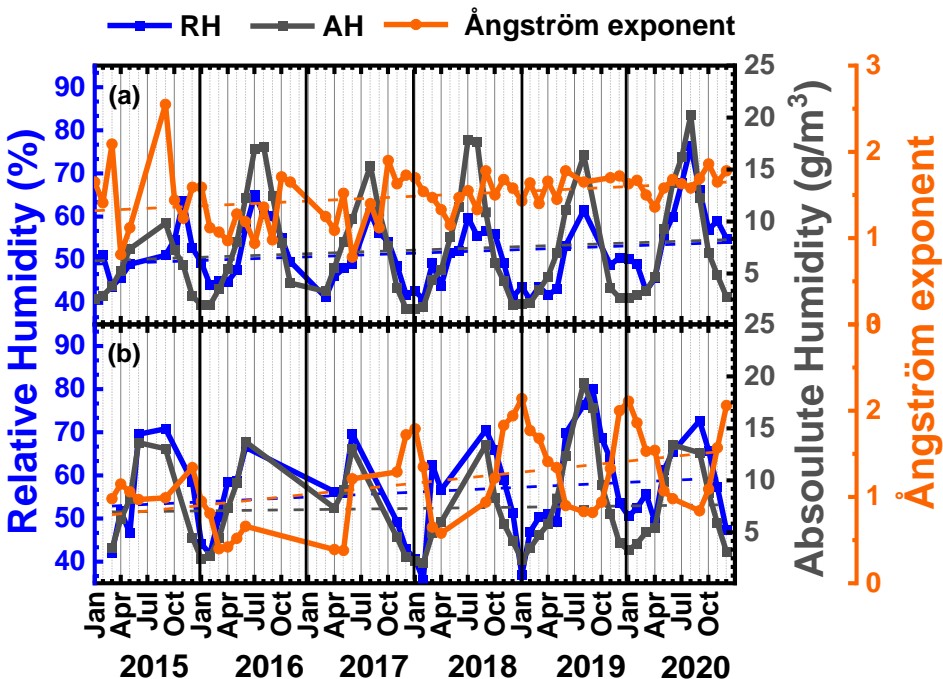

**Figure 4:** Monthly average Ångström exponent, relative and absolute humidity in (a) Seoul and (b) Ulsan.

**Table 3:** Mann–Kendall test and Slope for monthly averages of relative humidity, absolute humidity, and Ångström exponent.

| | | MK-test | | | Slope | |
|---|---|---|---|---|---|---|
| | | Trend | z | p | Simple linear regression | Sen's slope |
| Seoul | RH | No trend | +1.02 | 0.31 | +0.070 | +0.065 |
| | AH | No trend | +0.81 | 0.42 | +0.029 | +0.024 |
| | Ångström exponent | Increasing | +3.37 | 0.00 | +0.005 | +0.007 |
| Ulsan | RH | No trend | +1.19 | 0.23 | +0.095 | +0.174 |
| | AH | No trend | +0.45 | 0.65 | +0.010 | +0.019 |
| | Ångström exponent | Increasing | +2.49 | 0.01 | +0.011 | +0.015 |

RH, relative humidity; AH, absolute humidity

In this study, we determine that changes in the Ångström exponent, which imply changes in particle size, influence the increase in $PM_{2.5}$ MEE. Figure 5 shows the correlation between the Ångström exponent and both $PM_{2.5}$ MEE and $PM_{2.5-10}$ MEE in Seoul and Ulsan. Examining Figure 5, it is evident that the Ångström exponent in Seoul and Ulsan correlates with $PM_{2.5}$ MEE and $PM_{2.5-10}$ MEE. As demonstrated in both Figure 4 and Figure 5, an increase in the Ångström exponent in Seoul and Ulsan is associated with an increase in $PM_{2.5}$ MEE and a decrease in $PM_{2.5-}$

$_{10}$ MEE from 2015 to 2020. Figure 6 shows the correlation between the Relative humidity and both PM$_{2.5}$ MEE and PM$_{2.5-10}$ MEE in Seoul and Ulsan. In the case of relative humidity in Figure 6, it is confirmed that an increase in the value can affect the increase in PM$_{2.5-10}$ MEE. However, as confirmed in the MK-test in Table 3, no statistically significant trend for humidity can be found in both regions, and only the Ångström exponent in Ulsan and Seoul shows a significant increasing trend. It confirms that the increase in the Ångström exponent, rather than relative humidity, influences the light extinction characteristics of particles in Seoul and Ulsan.

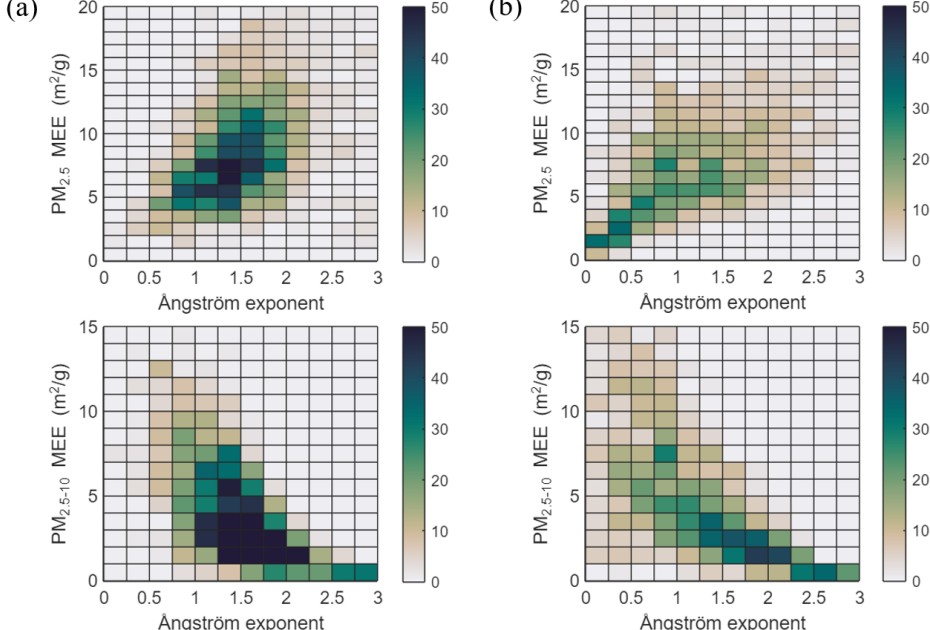

**Figure 5:** The correlation between the Ångström exponent and both PM$_{2.5}$ MEE and PM$_{2.5-10}$ MEE in (a) Seoul and (b) Ulsan.





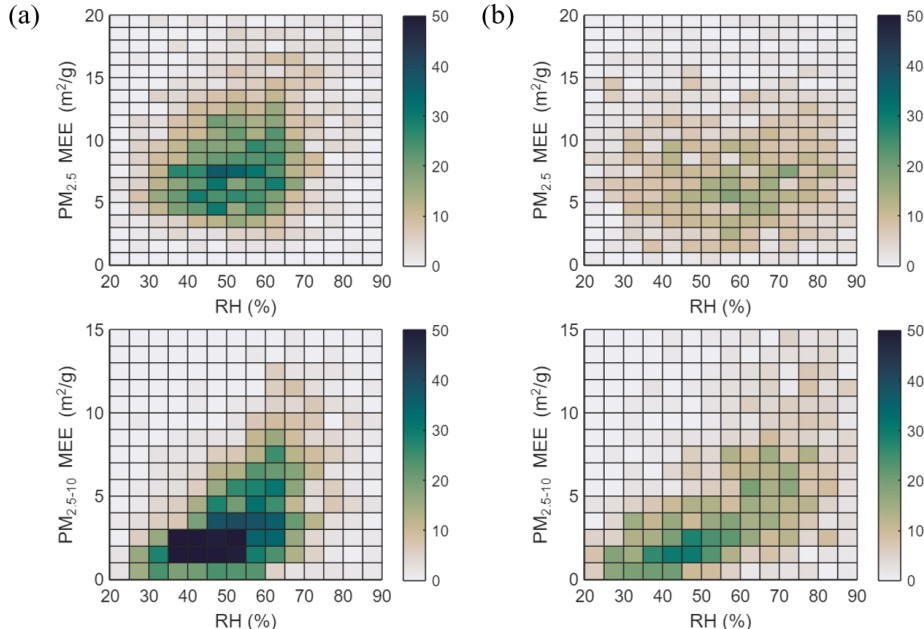

**Figure 6:** The correlation between the Relative humidity and both PM$_{2.5}$ MEE and PM$_{2.5-10}$ MEE in (a) Seoul and (b) Ulsan.

The Ångström exponent calculated in this study is the Ångström exponent of all atmospheric particles, including
both coarse- and fine-mode particles. Therefore, based on the Ångström exponent alone, it cannot be concluded
that the particle size of PM$_{2.5}$ is getting smaller over time. However, the high correlation between the Ångström
exponent and PM$_{2.5}$ MEE indicates that the size of PM$_{2.5}$ particles is decreasing, which is the most contributing
factor causing the increase in the Ångström exponent. The findings by Shin et al. (2023), who used AERONET
data and reported an increase in the Ångström exponent of fine-mode particles in major Northeast Asian cities
such as Seoul, Beijing, and Osaka, along with the results presented by Shin et al. (2022a), which revealed an
increase in the Ångström exponent of fine-mode particles in Northeast Asia from 2.57% in 2001 to 11.92% in
2018, corroborate the results of this study. Considering the results of previous studies, the increases in the
Ångström exponent and PM$_{2.5}$ MEE observed in this study are presumed to be caused by the decrease in the size
of fine-mode particles.

### 3.4 PM$_{2.5}$ MEE trend in Northeast Asia

There have been several cases in Northeast Asia wherein the improvement in visibility is not significant compared
with the reduction in PM$_{2.5}$ (Xu et al., 2020; Liu et al., 2020; Jeong et al., 2022). In this study, the overall PM$_{2.5}$
MEE trend in Northeast Asia is confirmed by comparing the PM$_{2.5}$ MEE values reported in studies conducted in
Northeast Asia with the annual trend of PM$_{2.5}$ MEE in Seoul and Ulsan, as reported in this study (Fig. 7). Except
for Jing et al. (2015), Zhang et al. (2022), Huang et al. (2024), Liu et al. (2020), and Shin et al. (2022b), all



previous studies reported the PM$_{2.5}$ MEE value at 550 nm wavelength. Jing et al. (2015) and Zhang et al. (2022) used 525 nm. Huang et al. (2024), Liu et al. (2020), and Shin et al. (2022b) used 520, 589, and 534 nm wavelengths, respectively. The increasing trend of PM$_{2.5}$ MEE at 532 nm, which is confirmed in this study, is consistent with

the results of previous studies. Using camera images, Shin et al. (2022b) reported high PM$_{2.5}$ MEE values (10.8 ± 6.9 m$^2 \cdot$g$^{-1}$) in Daejeon, South Korea, in 2021; these values were higher than the findings of previous studies conducted in Northeast Asia. Joo et al. (2022) examined the PM$_{2.5}$ MEE trends in eight regions of South Korea using visibility data; while their findings showed higher values than those observed in this study, the increasing trend of MEE is similar.

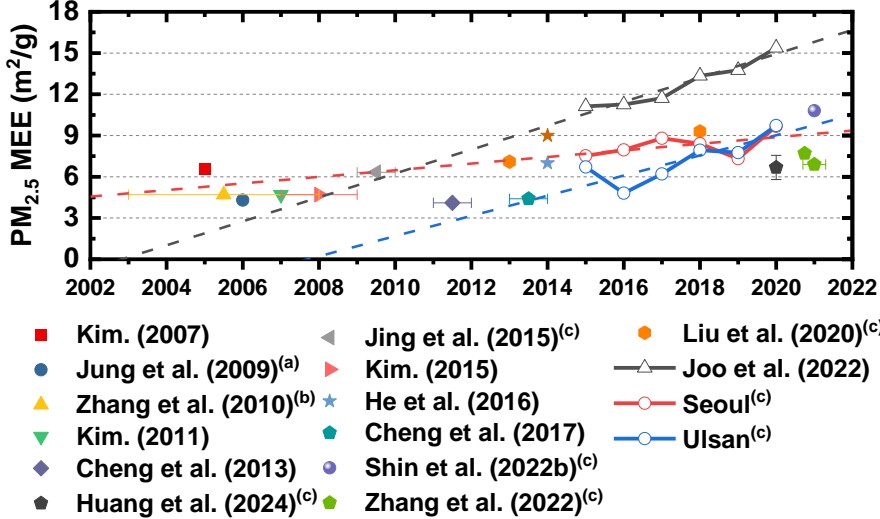

**Figure 7:** Comparison of PM$_{2.5}$ mass extinction efficiency (MEE) values from previous studies in Northeast Asia.

(a) Estimated by single scattering albedo of 0.8, according to the literature.

(b) The mass ratio of PM$_{2.5}$ to PM$_{10}$ was set to 0.56, and relative humidity was set to 40%, according to the literature.

(c) Studies using wavelengths other than 550 nm.


Liu et al. (2020) attributed the increase in PM$_{2.5}$ MEE in Eastern China to an increase in nitrate proportions and relative humidity. However, Joo et al. (2022), who used visibility data to calculate PM$_{2.5}$ MEE in South Korea, could not ascertain the exact causes of the increasing trend in MEE due to the lack of long-term information on particle size and composition. In this study, we utilize lidar data to calculate the Ångström exponent and assess

changes in particle size, thereby recognizing the possibility that these size changes contribute to the increase in PM$_{2.5}$ MEE in Seoul and Ulsan.

According to prior research conducted in Northeast Asia, it has been observed that while the contribution of primary PM$_{2.5}$ have decreased, the contribution of secondary PM$_{2.5}$ has increased compared with its contribution in the past (Lang et al., 2017; Du et al., 2020). As the proportion of the primary components of PM$_{2.5}$ has decreased,

and the proportion of smaller-sized secondary components has increased, the degree of light scattering by PM$_{2.5}$ particles may have increased even though PM$_{2.5}$ concentrations have decreased. In fact, the MEE of primary aerosols is lower than that of secondary aerosols (Wang et al., 2015; Du et al., 2022). The increase in the proportion



of secondary aerosol might have affected the hygroscopic growth of particles, thereby influencing MEE. Li et al. (2021), in their study on the chemical species of $PM_{2.5}$, confirmed that secondary pollutants, which are more prevalent in urban areas, cause greater hygroscopic growth. In contrast, primary combustion emissions were reported to exhibit less hygroscopic growth.

Additionally, changes in particle components that serve as precursors to secondary pollutants can also influence variations in light extinction. Jeong et al. (2022) reported that the minor improvement in visibility compared with the $PM_{2.5}$ mass concentration in Seoul from 2012 to 2018 could be attributed to the increase in ammonium nitrate ($NH_4NO_3$) due to a decrease in $SO_x$ and an increase in $NO_x$. An increase in $NO_x$, a precursor to secondary pollutants, can lead to an increase in $NH_4NO_3$. $NH_4NO_3$, compared with other components, has a greater degree of light extinction by particles, suggesting it may have contributed to the increase in MEE (Liu et al., 2019; Zhou et al., 2019; Jeong et al., 2022). Although a more extensive long-term analysis is required to understand the causes of the increase in $PM_{2.5}$ MEE in Northeast Asia, the increasing trend in $PM_{2.5}$ MEE and the enhanced light extinction by fine-mode particles are evident. Consequently, further research focusing not just on mass concentration but also on changes in particle size, composition and number concentration that can affect light extinction should be conducted in the future.



**4 Conclusion**

In this study, we utilized the AD-Net lidar data and PM mass concentrations to examine the trends in $PM_{2.5}$, $PM_{2.5–10}$, $PM_{10}$ MEE changes from 2015 to 2020 in Seoul and Ulsan. The results revealed that $PM_{2.5}$ MEE exhibited an increasing trend in Seoul [+0.04 ($m^2{\cdot}g^{-1}$)$mth^{-1}$] and Ulsan [+0.07 ($m^2{\cdot}g^{-1}$)$mth^{-1}$]. Meanwhile, $PM_{2.5–10}$ MEE exhibited a decreasing trend in Seoul [−0.02 ($m^2{\cdot}g^{-1}$)$mth^{-1}$] and Ulsan [−0.03 ($m^2{\cdot}g^{-1}$)$mth^{-1}$]. Therefore, in Seoul and Ulsan, the light extinction caused by coarse-mode particles ($PM_{2.5–10}$) has decreased, whereas the light

extinction caused by fine-mode particles ($PM_{2.5}$) has increased. To further identify the causes of these trends, we examined the trends in relative and absolute humidity as well as the changes in the Ångström exponent. The monthly average relative humidity exhibited increasing trends in Seoul (+0.070 %·$mth^{-1}$), Ulsan (+0.095 %·$mth^{-1}$). However, both relative humidity and absolute humidity were not statistically significant. The monthly average Ångström exponent significantly increased in Seoul (+0.005 $mth^{-1}$) and Ulsan (+0.011 $mth^{-1}$), suggesting that the

particles have become smaller. The increasing trend of the Ångström exponent, indicates that although the PM mass concentration has decreased, its effect on improving atmospheric visibility is negligible due to the increase in particles ($PM_{2.5}$) with smaller particle sizes. Additionally, by comparing the annual average $PM_{2.5}$ MEE in Seoul and Ulsan with the findings of previous studies conducted in Northeast Asia, it is confirmed that $PM_{2.5}$ MEE is generally increasing throughout Northeast Asia. Therefore, despite the consistent decrease in PM mass

concentration observed in previous studies across Northeast Asia, the limited improvement in visibility can be attributed to an increase in the number concentration of smaller particles. Thus, to enable citizens to noticeably perceive a reduction in PM and to lower their anxiety about air pollution, effective strategies that go beyond mere policies aimed at reducing PM mass concentration must be implemented. Therefore, it is essential to enact policies and conduct related research aimed at mitigating the increasing concentration of smaller secondary particles in

the atmosphere. However, in this study, we are unable to obtain long-term data on particle composition and, thus, could not determine the impact of compositional changes on MEE. Further research on MEE that considers changes in particle compositional characteristics is required.


**Author contributions**

Conceptualization and methodology: Y. Noh; Formal analysis: S. Joo and T. Kim; Writing—original draft: S. Joo and N. Dehkhoda; Manuscript review and editing: Y. Noh, M. Tesche, S. Joo, J. Shin and N. Dehkhoda; Simulation: J. Shin; All authors have read and agreed to the published version of the manuscript.


**Competing interests**

At least one of the (co-)authors is a member of the editorial board of Atmospheric Chemistry and Physics.

**Acknowledgements**

This work was supported by a grant of the "Graduate school of Particulate matter specialization" of the Korea Environmental Industry & Technology Institute grant, funded by the Ministry of Environment, Republic of Korea



and was supported by a grant (2023-MOIS-20024324) of Ministry-Cooperation R&D Program of Disaster-Safety funded by Ministry of Interior and Safety (MOIS, Korea).

**Data availability**

Lidar data were provided by courtesy of AD-Net (https://www.lidar.nies.go.jp/AD-Net). Lidar data for Seoul were provided by Seoul National University, under the supervision of Sang-Woo Kim (sangwookim@snu.ac.kr), and Lidar data for Ulsan were provided by the Ulsan National Institute of Science and Technology, under the supervision by Chang-Keun Song (cksong@unist.ac.kr).

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
