# Peer review of "Increased number concentrations of small particles explains perceived stagnation in air quality over Korea"

_EGUsphere, 2024_

## Author Comment (AC1)

[Figure]

**Figure 2:** Yearly and Monthly particulate matter (PM) mass concentration and visibility in (a) Seoul and (b) Ulsan.

[Figure]

**Figure 3:** Yearly and Monthly average mass extinction efficiency (MEE) in (a) Seoul and (b) Ulsan

[Figure]

**Figure 4:** Yearly and Monthly average Ångström exponent, relative and absolute humidity in (a) Seoul and (b) Ulsan.

---

## Author Response (AR1)

**Author's Response : Anonymous Referee #2**

Dear, referee

We would like to thank the referee for your valuable feedback and suggestions. Below are our responses to the comments provided.

**Comment 1:**

"One can regret that the root cause for the increase of the mass extinction efficiency is not found. This should be said explicitly in the abstract."

**Author's Response:**

Thank you for your valuable feedback. I will add the following sentence on lines 23-24: "However, due to the limitations in obtaining long-term composition data in this study, further research is needed to accurately determine the causes of the increase in $PM_{2.5}$ MEE."

**Comment 2:**

"The trends for several variables are given on a per month basis. Yet, the variables show a large seasonal cycle. Therefore, I believe that the per month basis is somewhat misleading and should be converted to a per year value."

**Author's Response:**

Thank you for your insightful comment. We agree that the seasonal variation complicates a straightforward by-eye inspection of the trends. Nevertheless, we have opted for this version because of data availability. As month-long data gaps exist in the data sets, the use of annual means would include an unbalanced representation of the results. We have now added annual mean values to the relevant plots, i.e., Figures 2, 3, and 4. These show trends that are in line with the ones inferred from using monthly means and help guiding the eye when inspecting the plots.

**Comment 3:**

"The introduction contains sentences such as "several citizens still believe that PM5 pollution remains serious." This clearly implies that the citizen are wrong in their fear of atmospheric pollution. Yet, although the pollution may have decrease, the levels are farm from harmless. Thus, the citizens are right in their perception that pollution remains serious"

**Author's Response:**

I agree with your comment that the sentence, "several citizens still believe that PM2.5 pollution remains serious." could imply that citizens are wrong in their concerns about atmospheric pollution. Therefore, to prevent any unintended interpretations, I will remove this sentence from the introduction.

**Comment 4:**

"Abstract However, both Relative humidity and Absolute humidity are not statistically significant". The trends of these variables are not statistically significant.

Line 233 : "Figure 2 and Table 1 show the monthly changes".   The figure show the monthly values, not the changes"

**Author's Response:**

We appreciate your careful review. We will revise the sentence in abstract and line 233.

<Abstract>

- However, both Relative humidity and Absolute humidity are not statistically significant

    → However, the trends of these variables are not statistically significant.

<Line 234>

- Figure 2 and Table 1 show the monthly changes in $PM_{2.5}$, $PM_{2.5-10}$, and $PM_{10}$ concentrations with the visibility across Seoul and Ulsan consistent with the findings of previous studies, the monthly PM mass concentrations in both cities are higher in winter and spring, and lower in summer and autumn

    → Figure 2 and Table 1 show the monthly values in $PM_{2.5}$, $PM_{2.5-10}$, and $PM_{10}$ concentrations with the visibility across Seoul and Ulsan consistent with the findings of previous studies, the monthly PM mass concentrations in both cities are higher in winter and spring, and lower in summer and autumn

We appreciate your thorough feedback and have implemented the suggested revisions to improve the clarity and accuracy of the manuscript.

**Author's Response : Anonymous Referee #1**

Dear, referee

We would like to thank the referee for your valuable feedback and suggestions. Below are our responses to the comments provided.

**Comment 1:**

""the particles become smaller each year". Clear mention smaller in size or number"

**Author's Response:**

Thank you for your valuable comment. We agree that the original phrasing could lead to ambiguity. To improve clarity, we have revised the sentence on line 23 to: "the size of the particles becomes smaller each year"

**Comment 2:**

"It is a very interesting work where the authors show how decreasing PM2.5 concentrations are not related to visibility. However, there is one confusion, the angstrom parameter values show increase in fine mode particles and the authors assumed e backscatter coefficients of fine mode are assumed to correspond to the light extinction caused by PM2.5. So when authors show increase of fine mode, does it also mean PM2.5 increases? on the contrary they also showed a decreasing trend in PM2.5."

**Author's Response:**

Thank you for your insightful comment and for highlighting this point of confusion. We understand the concern regarding the apparent contradiction between the increase in the fine mode particles indicated by the Ångström exponent and the decreasing trend in PM2.5 mass concentrations.

To clarify, the Ångström exponent is an indicator of the size distribution of aerosol particles, with higher values typically suggesting a dominance of fine particles. However, an increase in the Ångström exponent does not necessarily correspond to an increase in PM2.5 mass concentration. The observed decrease in PM2.5 concentration in our study reflects a reduction in the total mass of particulate matter with diameters less than 2.5 micrometers. The increase in the Ångström exponent observed in this study and the increase in the fine-mode Ångström exponent reported in previous studies suggest that the particle size within the fine mode has become smaller.

To make this point clearer, we have revised the manuscript to include the following statement on

page 16, line 390-393: "Considering the results of previous studies, the increases in the Ångström exponent and PM2.5 MEE observed in this study are presumed to be caused by the decrease in the size of fine-mode particles."

We hope this clarification resolves the confusion, and we appreciate your valuable feedback.

**Comment 3:**

"Moreover, the study would be well-rounded if composition information was added to know about hygroscopicity. The authors can look into satellite data for information about composition if ground data isn't available. Its not so straight forward to say organics are hydrophobic, one has to consider oxidised organics or SOAs"

**Author's Response:**

Thank you for your valuable feedback. We appreciate the suggestion to include composition information related to hygroscopicity and we agree that it is important to consider the role of composition.

Unfortunately, due to limitations in our data collection capabilities, we were unable to obtain detailed composition information for the aerosols studied. As a result, we could not directly address the hygroscopicity of specific components.

To improve clarity and avoid potential confusion, we have revised the manuscript to reflect a more accurate representation of hydrophilic and hydrophobic species. Specifically, on page 13, lines 325-326, the original text stated: "Hydrophilic species include secondary inorganic compounds (e.g., sulfate, nitrate, and ammonium) and sea salt, while hydrophobic species include organic carbon and black carbon." This has been modified to: "Hydrophilic species include secondary inorganic compounds (e.g., sulfate, nitrate, and ammonium) and sea salt, while hydrophobic species include dust and black carbon."

We hope this revision contributes to a clearer understanding of our study. We deeply appreciate your insightful comments and suggestions, which have been invaluable in enhancing the quality of our research.

---

## Author Response (AR3)

**Author's Response : Anonymous Referee # 1**

Dear, referee

We would like to thank the referee for your valuable feedback and suggestions. Below are our responses to the comments provided.

**Comment 1:**

"Abstract first line: "The atmospheric visibility in South Korea has not improved despite decreasing concentrations of particulate matter (PM)2.5." Please specify number or mass concentrations"

**Author's Response:**

We appreciate the reviewer's valuable suggestion. Accordingly, we have revised the first line of the abstract to read: "The atmospheric visibility in South Korea has not improved despite decreasing mass concentrations of particulate matter (PM)2.5."

**Comment 2:**

"Please include uncertainties in measurements of PM concentrations and subsequent calculation of rate of decrease"

**Author's Response:**

We sincerely appreciate your valuable suggestion regarding the inclusion of uncertainties in the PM concentration measurements and the subsequent calculation of the rate of decrease. The PM concentration data used in this study are finalized data validated through a rigorous process. After being measured according to official test methods, the data underwent a primary validation by the Seoul Metropolitan Air Quality Management Office and the Korea Environment Corporation, followed by a secondary validation conducted by the National Institute of Environmental Research (NIER), after which the final validated data were released. As such, specific uncertainty information for these data has not been provided, and we respectfully ask for your understanding that we are unable to offer additional details on this matter. However, given that the data are finalized by highly credible institutions, we have conducted our analysis based on the reliability of this data.

To address your concern and provide more context, we have added the following explanation to lines 102–108 of the manuscript:

"The PM concentration data are finalized and rigorously validated. The data were first measured according to official test methods. They then underwent primary validation by the Seoul Metropolitan Air Quality Management Office and the Korea Environment Corporation. This was followed by secondary validation performed by the National Institution of Environmental Research (NIER), after which the final validated data were released. Given that these data are validated and finalized by highly credible institutions, we have conducted our analysis based on their reliability."

We hope this addition clarifies our approach and addresses your concern. In addition, we used the Mann-Kendall (MK) test to assess uncertainties in the increasing and decreasing trends of PM, MEE, relative humidity, and Ångström exponent. This explanation is provided in lines 225-230, where z-scores and p-values are used to evaluate the extent of uncertainty associated with the calculated rates of increase or decrease. Thank you very much for your insightful feedback

**Comment 3:**

"Instead of looking at monthly trends, the authors could try to look at seasonal trends. It would be easier to attribute changes to sources/processes that are widespread across a season rather than a month"

**Author's Response:**

We sincerely appreciate the reviewer's insightful suggestion to examine seasonal trends for a clearer attribution of changes. In response, we conducted an analysis of seasonal trends in both Seoul and Ulsan. However, in both locations, we found that the seasonal trends did not show significant differences from the monthly trends already presented in the manuscript. Additionally, due to data limitations in certain months, particularly in Ulsan, seasonal trends could not be adequately represented, making it less suitable for this approach.

To clarify this in the manuscript and reduce potential confusion, we have added the following sentences to lines 295–297:

"Furthermore, we conducted an analysis of seasonal trends in both Seoul and Ulsan. However, the seasonal trends did not reveal any significant differences compared to the monthly trends already presented in this study."

By including this explanation, we aim to enhance the clarity of the manuscript while maintaining a focused presentation of the results. We are grateful for the opportunity to further validate our findings and address the reviewer's valuable suggestion.

**Comment 4:**

"It is very evident from an aerosol size distribution that the smaller particles have a negligent impact on mass of the aerosols. Isnt the conclusion of the study very obvious or similar to the already known facts. How is this study contributing in understanding something new or the gaps of the knowledge we already have? This study feels like use of multiple statistical approaches to come round an already known fact that smaller size aerosols donot contribute to mass distributionThe study needs more scientific backing."

**Author's Response:**

Thank you for your valuable question regarding the scientific contribution of this study. While it is widely recognized that smaller aerosol particles contribute minimally to the total aerosol mass, our study brings new insights by quantifying the impact of particle size on mass extinction efficiency (MEE) specifically in the context of Northeast Asia, with a focus on South Korea. The increase in PM2.5 MEE despite reductions in PM2.5 mass concentrations highlights a critical issue: that smaller particles, although contributing little to mass, significantly affect visibility due to their high scattering efficiency.

Our study also addresses an important gap in the current understanding by using lidar data to examine the trends in PM2.5 MEE, relative humidity, and the Ångström exponent, which is indicative of particle size. These analyses allow us to attribute the observed trends in MEE to reductions in particle size rather than to overall mass concentration. This approach provides a detailed understanding of visibility trends, which have not shown improvements proportional to the reductions in PM mass concentrations. These findings emphasize the need for policy measures targeting not only mass reduction but also the composition and size of particles to improve air quality and visibility effectively.

We believe that this approach strengthens the scientific understanding of the optical properties of aerosols in the region, contributing valuable knowledge on how fine-mode particles influence public perception and environmental policies. We sincerely appreciate your insightful question regarding the scientific contribution of this study. The contributions and implications of our findings, including the focus on the optical properties of aerosols and their impact on visibility and policy-making, have been comprehensively addressed in the conclusion section of the manuscript. As such, we kindly request the reviewer to revisit this section for further clarification.

**Comment 5:**

"It is interesting to note that the authors attribute increase in relative humidity to increase in MEE due to hygroscopic growth. My question is: doesn't hygroscopic growth of particles increase the size and eventually contribute to mass? Again the authors mention a reduction in particle size result in larger MEE. It's a bit confusing if the hygroscopic growth is increasing MEE or the reduction in particle size"

**Author's Response:**

Thank you for your insightful question regarding the impact of hygroscopic growth and particle size reduction on mass extinction efficiency (MEE). To clarify, both hygroscopic growth and smaller particle sizes contribute to higher MEE, but they do so through slightly different mechanisms.

When relative humidity increases, hygroscopic particles absorb moisture, causing them to grow in size. This growth leads to an increase in their scattering cross-section, meaning they scatter more light without a significant increase in their dry mass. As a result, MEE increases because more light is scattered per unit of particle mass under humid conditions.

Similarly, when particle sizes are intrinsically smaller, the overall scattering cross-section in the same mass concentration is larger compared to having a few large particles. This is because a collection of smaller particles, with a greater surface area-to-mass ratio, scatters light more effectively than a single larger particle of the same total mass, leading to a higher MEE.

In summary, MEE increases both when particles grow due to hygroscopic effects (under higher humidity) and when particles are smaller in dry conditions, as both situations enhance the scattering cross-section relative to the mass concentration. We hope this explanation addresses your question and clarifies the mechanisms by which both hygroscopic growth and particle size reduction impact MEE.

Additionally, we have included the following statement in our paper on lines 326-328 to further clarify this point: " Both hygroscopic growth, which increases particle size and scattering cross-section under high humidity, and the presence of smaller particles, which scatter light more efficiently per unit mass, contribute to the increase in MEE."

**Comment 6:**

"Why is the monthly trend of relative humidity increasing? Can you associate it with any atmospheric/anthropogenic process happening in that region?"

**Author's Response:**

Thank you for your thoughtful question regarding the trend of relative humidity observed in our study. Upon careful review, our findings indicate that the monthly trends in relative humidity do not exhibit a statistically significant increase. The results from the Mann-Kendall test, along with the associated p-values, do not support a conclusive upward trend. While our analysis reveals some slight month-to-month fluctuations, these variations do not constitute a clear or statistically supported long-term trend. As for potential atmospheric or anthropogenic processes that could influence relative humidity, we recognize that factors such as temperature changes, local and regional weather patterns, and urbanization can play a role. However, given the lack of a statistically significant trend in our data, we are cautious about drawing speculative associations with these factors. We sincerely appreciate your attention to this detail.

We appreciate your thorough feedback and have implemented the suggested revisions to improve the clarity and accuracy of the manuscript.